# Longitudinal Follow-Up of Serum and Urine Biomarkers Indicative of COVID-19-Associated Acute Kidney Injury: Diagnostic and Prognostic Impacts

**DOI:** 10.3390/ijms242216495

**Published:** 2023-11-18

**Authors:** Yahya Lablad, Charlotte Vanhomwegen, Eric De Prez, Marie-Hélène Antoine, Sania Hasan, Thomas Baudoux, Joëlle Nortier

**Affiliations:** 1Laboratory of Experimental Nephrology, Faculty of Medicine, Université Libre de Bruxelles, Erasme Campus, 808 Route de Lennik, 1070 Brussels, Belgium; charlotte.vanhomwegen@ulb.be (C.V.); eric.de.prez@ulb.be (E.D.P.); marie-helene.antoine@ulb.be (M.-H.A.); sania.hasan@ulb.be (S.H.); thomas.baudoux@hubruxelles.be (T.B.); 2Department of Nephrology, Dialysis and Renal Transplantation, Erasme University Hospital, Erasme Campus, 1070 Brussels, Belgium

**Keywords:** AKI, COVID-19, biomarkers, NGAL, LAP, CCL14, cystatin C, suPAR

## Abstract

In patients hospitalized for severe COVID-19, the incidence of acute kidney injury (AKI) is approximately 40%. To predict and understand the implications of this complication, various blood and urine biomarkers have been proposed, including neutrophil gelatinase-associated lipocalin (NGAL), chemokine (C-C motif) ligand 14 (CCL14), cystatin C, leucine aminopeptidase (LAP), and soluble urokinase plasminogen activator (suPAR). This study, conducted between mid-January and early May 2021, aimed to assess the diagnostic and prognostic capabilities of these biomarkers in a cohort of COVID-19 patients monitored during the initial two weeks of hospitalization. Among the 116 patients included in this study, 48 developed AKI within the first three days of hospitalization (41%), with 29 requiring intensive care unit (ICU) admission, and the overall mortality rate was 18%. AKI patients exhibited a statistically significant increase in urinary LAP levels, indicating acute tubular injury as a potential mechanism underlying COVID-19-related renal damage. Conversely, urinary NGAL and CCL-14 excretion rates did not differ significantly between the AKI and non-AKI groups. Importantly, elevated plasma suPAR and cystatin C levels upon admission persisted throughout the first week of hospitalization and were associated with unfavorable outcomes, such as prolonged ICU stays and increased mortality, irrespective of AKI development. In conclusion, this study underscores the early predictive value of urinary LAP levels in identifying acute tubular injury in COVID-19-induced AKI. Moreover, elevated plasma suPAR and cystatin C levels serve as valuable prognostic markers, offering insights into the short-term morbidity and mortality risks among COVID-19 patients, regardless of AKI occurrence. These findings shed light on the complex interplay between COVID-19, renal injury, and biomarkers with diagnostic and prognostic potential.

## 1. Introduction

In December 2019, the pandemic associated with the SARS-CoV-2 coronavirus spread rapidly around the world. Although the disease associated with this coronavirus (COVID-19) corresponds primarily to severe acute respiratory syndrome, other manifestations have been observed, leading notably to cardiovascular disease and renal impairment [1]. Acute kidney injury (AKI) is a frequent complication in patients hospitalized with a severe form of COVID-19. Recent studies have suggested a high prevalence of AKI at around 40% [2,3,4], varying according to patients’ comorbidities and risk factors [5]. AKI is also associated with other severe complications, high medical costs, and prolonged hospital stays. It is also associated with an increased risk of death and the development of chronic kidney disease (CKD) [6,7,8].

The pathogenesis of AKI in patients with COVID-19 is not entirely clear. Multiple predisposing factors appear to be involved, such as obesity, diabetes, and arterial hypertension [9]. Among the most frequently cited pathophysiological mechanisms is an exaggerated immune response in some patients known as the “cytokine storm”. This immunological disorder can lead to systemic inflammation affecting the kidneys and other organs [10]. In addition, tissue hypoxia that is common in COVID-19 patients can cause kidney damage via vasoconstriction leading to organ ischemia, and thrombotic microangiopathy lesions have also been reported [11].

There are also reports of the involvement of the SARS-CoV-2 virus itself. Indeed, the interaction between the SARS-CoV-2 virus and ACE-2 can lead to inflammation and extensive tissue damage [12,13], which can directly damage renal tubular cells by binding to angiotensin-converting enzyme 2 (ACE-2). However, the direct role of the virus as indicated by its presence in kidney cells remains controversial.

Two functional parameters, serum creatinine (Cr) and urine output, are traditionally used to define the severity of AKI, but their predictive value is limited by the fact that they change late in relation to the onset of renal damage, resulting in low sensitivity and specificity. In addition, they may be affected by factors such as patients’ age, sex, muscle mass, and hydration status [14].

Several biomarkers have been developed to improve the early detection of AKI. These biomarkers attempt to target different segments of the nephron. Some have been validated in a variety of clinical settings. The most widely studied to date are cystatin C, neutrophil gelatinase-associated lipocalin (NGAL), leukocyte alkaline phosphatase (LAP), kidney injury molecule-1 (KIM-1) and, more recently, CCL14 [14,15].

The aims of our study are (1) to identify clinical and biological factors that are predictive of AKI in patients with severe COVID-19 disease hospitalized at Erasme University Hospital in the Emergency Department, and (2) to evaluate the contribution of specific serum and urinary biomarkers in characterizing renal impairment in these patients and predicting their clinical prognosis.

## 2. Results

### 2.1. Clinical Characteristics of Patients at Inclusion

We included a total of 116 patients in our study. The mean age of the cohort was 61 years (ranging from 18 to 85 years), and the M/F ratio was 1.65. Their mean BMI was 30, with 90 patients (78%) having a BMI > 25.

Among the patients included, 54 (46.6%) were hospitalized in the ICU and 62 (53.5%) in the COVID Unit, based on the referrals from the Emergency Department according to the severity of their respiratory illness. The total number of deaths in our cohort was 21 (18.1%).

With regard to comorbidities, 36 patients (31%) were smokers, 36 patients (31%) suffered from type 2 diabetes, and 69 patients (59.5%) suffered from hypertension. Preexisting CKD was recorded in 15 patients (12.9%), with 3 in stage 1, 6 in stage 2, and 6 in stage 3 (Table 1).

### 2.2. Incidence of AKI in the Study Cohort

A total of 48 patients developed AKI within the first 3 days of hospitalization (41%): 21 at AKIN stage 1 (44%) and 27 at AKIN stage 3 (56%), of whom 3 required transient hemodialysis (Figure 1).

### 2.3. Clinical Factors and Biological Parameters Associated with Early-Onset AKI in COVID-19 Patients

In the univariate analysis, the results indicated that patients with hypertension or BMI > 30 were significantly more likely to develop AKI. In addition, a significantly higher rate of ICU admission was observed in patients with AKI (Table 2).

Table 2 also shows the biological parameters stratified according to the presence or absence of AKI within the first 3 days of hospitalization, with statistically significant differences found on Day 0 for urea, creatininemia, urinary creatinine, and CRP and on Day 3 for urea, AST, and urinary white blood cells, respectively. All data mentioned were significantly higher in AKI patients.

### 2.4. AKI Predictive Biomarkers

#### 2.4.1. Time Course of Urinary LAP and CCL14 According to the Presence or Absence of AKI

During hospital stay, the median urinary LAP levels were significantly higher in AKI patients compared to non-AKI patients from the third day of hospitalization (Figure 2).

AKI patients tended to have higher urinary levels of CCL14 upon admission and on the third day of hospitalization, but no statistically significant difference was found between groups (Figure 3).

#### 2.4.2. Time Course of Plasma and Urinary NGAL Levels According to the Presence or Absence of AKI

Upon admission and during hospitalization, the median plasma and urinary NGAL levels tended to be higher in AKI patients than in non-AKI patients, but no statistically significant difference was found.

### 2.5. Morbi-Mortality Prognosis Biomarkers

#### 2.5.1. Time Course of Cystatin C Plasma Levels

Upon admission and at every time point of measurement during hospitalization, plasma cystatin C levels were significantly higher in patients who died during hospitalization (Figure 4).

#### 2.5.2. Time Course of Plasma suPAR Levels According to the Severity of Clinical Condition

From Day 0 to Day 7, patients admitted to the ICU had significantly higher plasma suPAR levels than patients hospitalized in the COVID Unit (Figure 5).

#### 2.5.3. Time Course of suPAR Plasma Levels According to Patients’ Life Prognosis

Upon admission and at every time point of sample collection during hospitalization, plasma suPAR levels were significantly higher in patients who finally deteriorated and died (Figure 6).

## 3. Discussion

In our study, the incidence of AKI associated with COVID-19 was 41%, a result that was quite similar to those found in the literature. A study carried out in the USA involving almost 4000 patients reported an incidence of 46% [16]. The incidence of AKI in patients admitted to conventional hospitals was 31%, while in ICUs, the incidence was 60%. Similar results were reported in a study carried out in both the USA and Australia [17].

Several clinical and biological factors found in our patients were associated with the occurrence of AKI, as reported in the literature, such as arterial hypertension [17] and BMI > 30 [18].

Other similarities with the literature were found in our AKI patients: higher serum urea and creatinine levels on Day 0 [19], higher CRP levels [20], and higher ASAT levels [21]. The diagnosis of AKI in our study was primarily determined by assessing creatinine levels and urine output. Notably, the creatinine levels did not exhibit statistically significant difference on Day 3. This lack of statistically significant difference could be attributed to the rapid development of anuria among AKI patients, coupled with the administration of substantial volumes of saline serum infusion, which contributed to a certain degree of hemodilution. Consequently, the diagnosis of AKI in our study was chiefly established based on the presence of oligo- and anuria, in alignment with the criteria outlined by the Acute Kidney Injury Network (AKIN).

Our study revealed that patients afflicted with acute kidney injury (AKI) exhibited a median proteinuria level of 0.59 [0.13, 1.69], while those without AKI had a median of 0.44 [0.02, 1.14]. Notably, the calculated *p*-value of 0.425 suggests that there was no discernible statistical dissimilarity in proteinuria levels between these two patient groups. The elevated proteinuria levels, irrespective of the development of AKI, might be plausibly attributed to the influence of COVID-19, as elucidated in a comprehensive literature review from 2021 [9].

The physio-pathological hypothesis of acute tubular damage in relation to the occurrence of AKI complicating a severe COVID-19 infection was tested by using the biomarkers NGAL (blood and urine), CCL 14 (urine), and LAP (urine). Only the last biomarker was found to be statistically increased in patients with AKI, reflecting a massive urinary release at the apical brush border of the proximal tubular epithelium. No statistically significant increase in NGAL and CCL-14 could be found in patients with AKI, probably because of the wide dispersion of the values obtained.

Regarding plasma NGAL, our results do not concur with those reported in 2021, in which the authors were able to measure significantly higher blood levels as early as Day 0 in severe COVID-19 patients who were hospitalized in the ICU and developed AKI [22]. Similarly, another study that was also carried out in 2021 reported that high levels of urinary NGAL were associated with the severity of kidney damage and a poor prognosis in COVID-19 patients [23]. These differences can be explained by the size of the cohorts in the two studies mentioned.

Regarding urinary CCL14, the RUBY study, published in two parts (2020 and 2022), reported the value of this biomarker for early detection of AKI and prediction of pejorative progression to CKD [15,24]. Again, these differences could be explained by the size of the cohort. Of note, the systematic administration of corticosteroids in our patient cohort could have interfered with local inflammatory processes and, hence, reduced the CCL14 urinary levels detected.

Interestingly, we were able to measure a compelling and statistically significant increase in the urinary excretion rate of LAP among patients who developed AKI over the duration of this study. This enzyme is located at the brush border of the proximal tubule, suggesting a notable injury of this nephron segment in the context of a COVID-19 infection. In the context of a COVID-19 infection, the heightened immune response often leads to an excessive and dysregulated release of neutrophils into the bloodstream. These activated neutrophils play a pivotal role in the immune defense against the virus, yet their hyperactivity may inadvertently contribute to collateral tissue damage. This phenomenon is particularly relevant to the kidneys as excessive inflammation can compromise renal function and integrity [9].

Regarding the potential predictive value of suPAR plasma levels, no statistically significant difference was found between AKI and non-AKI patients. Of interest, patients admitted to the ICU had significantly higher plasma SuPAR levels than those hospitalized in the COVID Unit. In addition, patients who died during the study period had significantly higher plasma suPAR levels than surviving patients from admission to Day 7. These results confirm the value of this biomarker as a predictive parameter of the severity of the clinical course in the days following admission and, therefore, a predictor of vital prognosis [25]. The increase in plasma suPAR, reflecting the inflammation and immune response described in severe COVID-19, reinforces the theory of the systemic inflammatory response known as the “cytokine storm” induced by the virus, leading to multiorgan damage [26]. In the same line, patients who died during the study had significantly higher plasma cystatin C levels than surviving patients from admission to Day 14. These findings validate the utility of this biomarker as a predictive parameter for gauging the severity of the clinical progression during hospitalization. Furthermore, they underscore its pivotal role in forecasting vital prognosis, aligning with the observations made in a 2020 study conducted in China [27].

## 4. Materials and Methods

### 4.1. Data Collection

This is a prospective observational study conducted at Erasme University Hospital during the third wave of the SARS-CoV-2 pandemic between February and May 2021.

We included patients over 18 years of age who were admitted to a COVID inpatient unit or an intensive care unit (ICU) with the diagnosis of SARS-CoV-2 infection confirmed upon admission via a PCR test performed on a naso- or oro-pharyngeal sample.

Only patients for whom a blood sample was taken within 48 h of admission were included. All patients who were able to understand the information document and signed the informed consent form were included. Renal transplant recipients and dialysis patients were excluded from the study.

### 4.2. Study Variables

Exhaustive data collection was carried out, including the following:

Clinical data upon admission: age, gender, weight, height, body mass index (BMI), smoking, blood pressure, heart rate, comorbidities, and current drug medications.

Biological data: whole blood count, C-reactive protein (CRP), liver enzymology, urea and creatinine levels, and urine spot.

Specific biomarkers: plasma and urine NGAL, plasma-soluble urokinase plasminogen activator receptor (suPAR), urine CCL14, urine leukocyte alkaline phosphatase (LAP), and plasma cystatin C.

All clinical and biological data, as well as specific biomarker measurements, were obtained on the day of patient admission (Day 0), Day 3, Day 5, Day 7, and Day 14. An additional sample was collected during the ambulatory period from hospital discharge to 2–6 months post-discharge.

A total of 116 patients met the inclusion criteria. The primary endpoint was the occurrence of AKI between Day 0 and Day 3 as defined according to the AKIN criteria [28].

### 4.3. Selection and Assay of Specific Biomarkers

The selection of specific biomarkers for the detection of AKI depends on several factors, including the sensitivity and specificity of the tests used, their commercial availability, their costs, and their predictive characteristics regarding the early detection of AKI and the presumed underlying pathophysiology of AKI.

At the time of this study, given the pathophysiological hypotheses regarding COVID-19-related renal damage, our choice of biomarkers was guided by the 2020 Acute Disease Quality Consensus Conference, with NGAL, LAP, CCL14, and cystatin C being proposed for their ability to detect structural damage at an early stage and predict renal prognosis [29].

NGAL was measured using an ELISA test (Human Lipocalin-2/NGAL DuoSet ELISA) (R&D Systems, Inc., Minneapolis, MN, USA). and performed in the laboratory of experimental nephrology. A blood assay was performed in addition to a urine assay (results related to urine creatinine).

Urinary CCL14 was measured using an ELISA test (Human CCL14a/HCC-1 DuoSet ELISA) (R&D Systems, Inc., Minneapolis, MN, USA), and each result was related to urinary creatinine.

Urinary LAP was measured using a spectrofluorimetric assay with a fluorescent Leu-AMC substrate sold by Sigma (ref: L2145) (Merck Life Science A/S, Søborg, Denmark). After incubation with the samples, the fluorescence of the released AMC was measured (excitation at 367 nm and emission at 440 nm), the concentration was calculated using a LAP standard curve (sigma ref: L5006) (Merck Life Science A/S, Søborg, Denmark, and each result was related to urinary creatinine.

Plasma suPAR assay was proposed in 2022 as a “triage” prognostic marker for patients with suspected COVID-19 upon arrival at an emergency room. It was measured using a laboratory ELISA test (suPARnostic^®^ AUTO Flex ELISA, Virogate A/S, Birkeroed, Denmark).

Plasma cystatin C was measured using an Duoset Elisa sold by RnDSystems (ref: DY1196 and DY008B) (R&D Systems, Inc., Minneapolis, MN, USA).

### 4.4. Statistical Analyses

All data were analyzed using the R statistical software version 1.3.1093.

Descriptive statistics were presented as median (25th percentile and 75th percentile) for continuous variables and as absolute number (percentage) for qualitative data. Comparisons between the AKI and non-AKI groups were performed using Fisher’s test and Wilcoxon–Mann–Whitney test for categorical and continuous variables, respectively. The significance threshold was set at *p* < 0.05.

## 5. Conclusions

Our findings provide valuable insights into the potential association between elevated LAP levels and the development of AKI in COVID-19 patients. While further research is necessary to fully elucidate the precise mechanisms underlying this relationship, our observations underscore the significance of monitoring LAP as a potential diagnostic and prognostic tool in the context of AKI after a COVID-19 infection. On the other hand, elevated plasma suPAR and cystatin C levels upon admission and persistent into early hospitalization are predictive of significant short-term morbidity and mortality, irrespective of the occurrence of AKI.

## Figures and Tables

**Figure 1 ijms-24-16495-f001:**
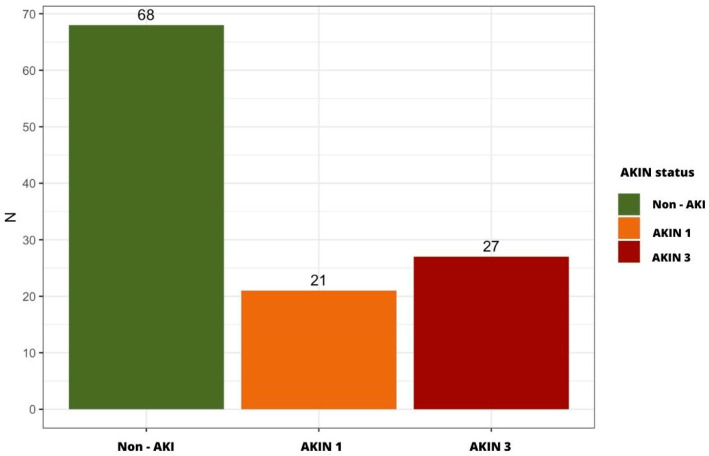
Distribution of patients in the cohort according to the occurrence or non-occurrence of AKI within the first 3 days of hospitalization (AKIN stages 1 vs. 3).

**Figure 2 ijms-24-16495-f002:**
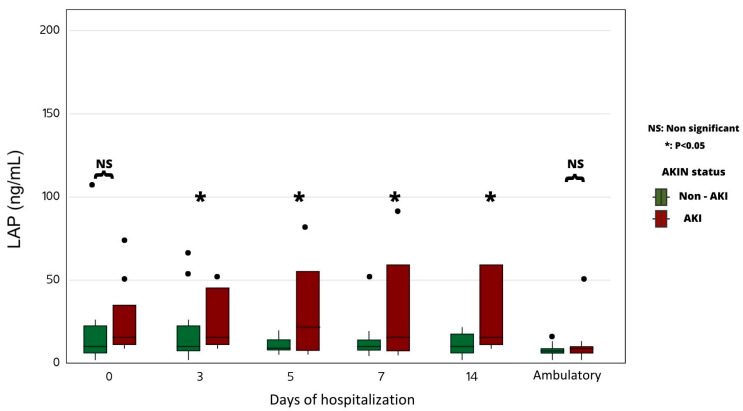
Time course of urinary LAP levels in AKI vs. non-AKI groups.

**Figure 3 ijms-24-16495-f003:**
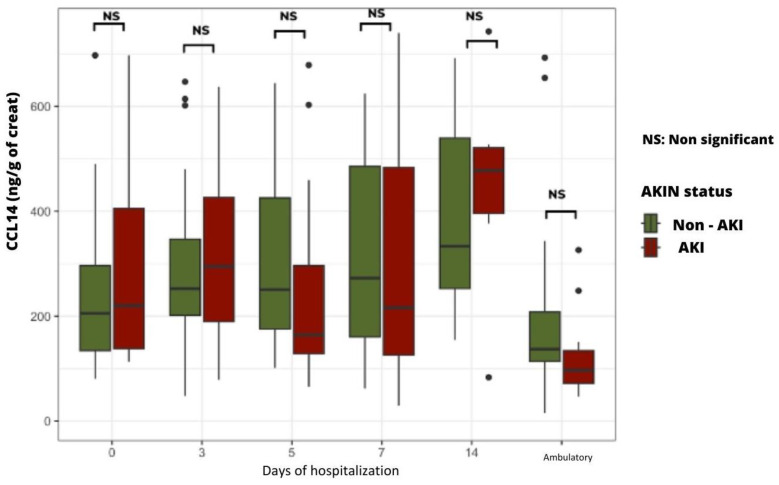
Time course of urinary CCL14 levels in AKI vs. non-AKI groups.

**Figure 4 ijms-24-16495-f004:**
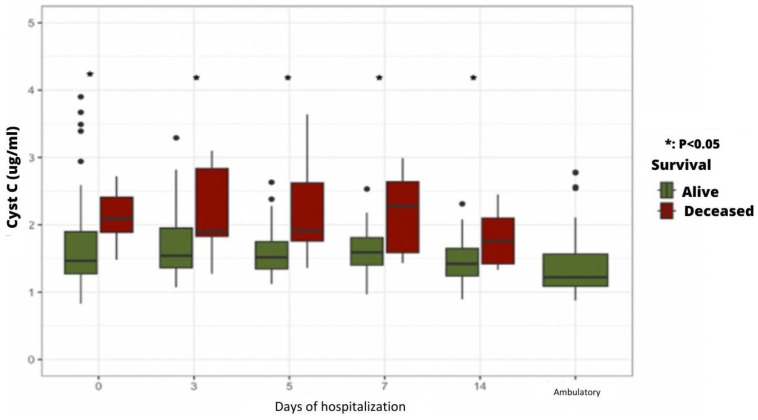
Time course of plasma cystatin C levels according to patient death or survival.

**Figure 5 ijms-24-16495-f005:**
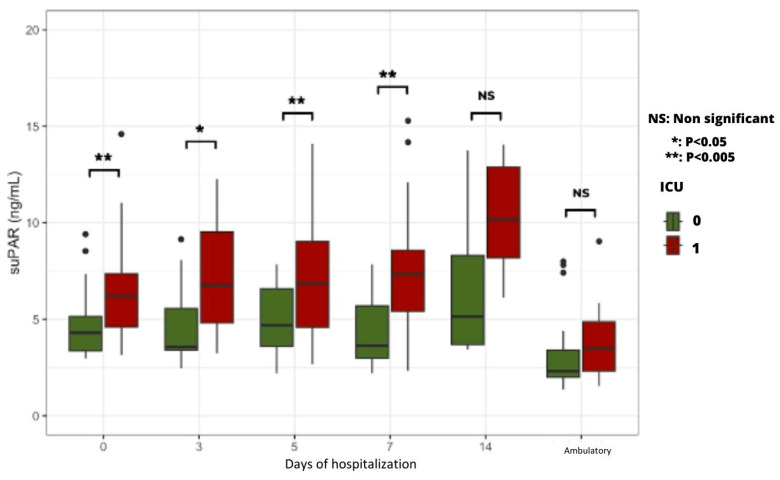
Time course of plasma suPAR levels according to ICU vs. COVID Unit hospitalization.

**Figure 6 ijms-24-16495-f006:**
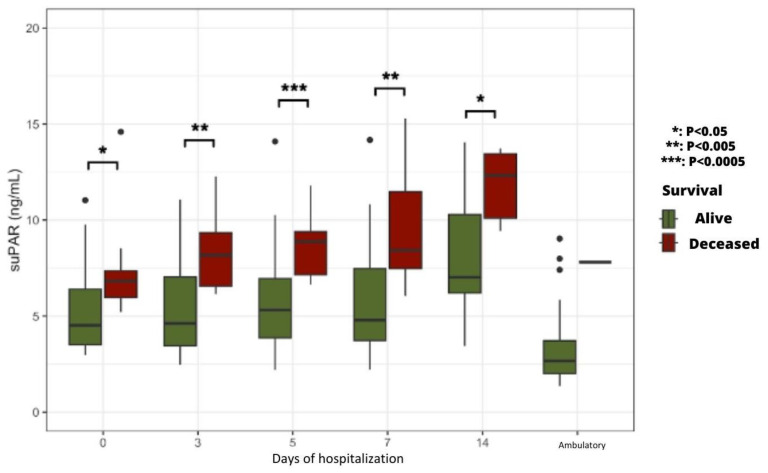
Time course of plasma suPAR levels in the deceased vs. surviving patient groups.

**Table 1 ijms-24-16495-t001:** Clinical characteristics of the included patients.

Characteristics	Data (N = 116)
Gender Female Male	44 (37.9%) 72 (62.1%)
Average age (years)	60.7 (from 18 to 85)
Active smoking	36 (31%)
Average BMI Overweight (25–29.9) Moderate obesity (30–34.9) Severe or major obesity (>35)	30 (17 to 53) 41 (35%) 29 (25%) 20 (17%)
ICU admission	54 (46.6%)
Deaths	21 (18.1%)
Comorbidities Type 2 diabetes HTA CKD Neurological disease Rheumatological disease Neoplasia status Pulmonary disease	36 (31%) 69 (59.5%) 15 (12.9%) 10 (8.6%) 12 (10.3%) 14 (12%) 18 (15.5%)

**Table 2 ijms-24-16495-t002:** Clinical characteristics and biological parameters stratified according to the occurrence of AKI.

Characteristics and Parameters	Non-AKI (N = 68)	AKI (N = 48)	*p*-Value
Gender Female Male	29 (43%) 39 (57%)	15 (31%) 33 (69%)	0.25
Age <65 years ≥65 years	38 (56%) 30 (44%)	29 (60%) 19 (40%)	0.7
Active smoking	19 (28%)	16 (33%)	0.54
Deaths	11 (16%)	10 (21%)	0.47
BMI > 30	18 (26%)	32 (67%)	<0.05
ICU admission	24 (35%)	29 (60%)	<0.05
Comorbidities Hypertension Type 2 diabetes CKD Neurological disease Rheumatology disease Neoplasia status Pulmonary disease	33 (49%) 19 (28%) 8 (12%) 5 (7%) 9 (13%) 7 (10%) 9 (13%)	35 (73%) 17 (35%) 6 (12%) 4 (8%) 3 (6%) 6 (12%) 9 (19%)	<0.05 0.42 1 1 0.36 0.77 0.44
Day 0 Hemoglobin (g/dL) White blood cells Platelets CRP (mg/L) ASAT (UI/L) Urea (mg/dL) Creatininemia (mg/dL) Urinary creatinine (mmol/L) Proteinuria (mg/L)	13 [11, 14] 7.0 [4.9, 10] 240 [180, 340] 70 [26, 110] 39 [26, 52] 33 [25, 44] 0.79 [0.66, 0.99] 120 [64, 160] 0.44 [0.02, 1.14]	13 [12, 14] 6.8 [5.1, 8.3] 230 [160, 310] 110 [51, 190] 37 [27, 52] 44 [31, 63] 0.93 [0.72, 1.1] 140 [110, 200] 0.59 [0.13, 1.69]	0.75 0.36 0.21 <0.05 0.37 <0.05 <0.05 <0.05 0.425
Day 3 ASAT (UI/L) Urea (mg/dL) Creatininemia (mg/dL) Urinary creatinine (mmol/L)	27 [21, 44] 34 [29, 49] 0.73 [0.59, 0.87] 84 [54, 140]	37 [27, 59] 46 [35, 65] 0.73 [0.63, 1.1] 110 [86, 140]	<0.05 <0.05 0.214 0.108

All data presented as median (25th percentile and 75th percentile).

## Data Availability

The data presented in this study are available from the corresponding author upon request.

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
