# Peer review of "Longitudinal Follow-Up of Serum and Urine Biomarkers Indicative of COVID-19-Associated Acute Kidney Injury: Diagnostic and Prognostic Impacts"

_ijms, 2023, doi:10.3390/ijms242216495_

Round 1

Reviewer 1 Report

Comments and Suggestions for Authors

The authors' research reports very interesting results on analyzing biomarkers associated with AKI in COVID-19 patients, which can help detect AKI early.

Please arrange Figure 1 in proportion.

It would be essential to provide more detailed data regarding the age group under 65 in Table 1, as this would be crucial for assessing the risk associated with age.

It seems necessary to examine the relationship between viral load in SARS-CoV-2 samples and COVID-19 patients. Since the viral load in samples is determined as a result of the replication process in the human body after virus infection and within the patient's organs, this relationship is important. If you have data on the SARS-CoV-2 results from samples for each patient, please include them. If not, consider reviewing the relevant literature and discussing this in the manuscript.

Author Response

The authors' research reports very interesting results on analyzing biomarkers associated with AKI in COVID-19 patients, which can help detect AKI early.

Please arrange Figure 1 in proportion 

This has been done in the revised version of the manuscript.

It would be essential to provide more detailed data regarding the age group under 65 in Table 1, as this would be crucial for assessing the risk associated with age : 

We thank the reviewer for their valuable feedback and wish to respond to the concerns raised regarding the age category. As explicitly stated in our manuscript, our initial findings indicated that age did not serve as a predictive factor for the onset of AKI. However, we acknowledge the importance of assessing the risks specifically within the under-65 age category. Regrettably, for this first study, examining this subgroup was not within the scope of our research objectives. In a subsequent phase of our investigation, we will ensure to address this aspect.

It seems necessary to examine the relationship between viral load in SARS-CoV-2 samples and COVID-19 patients. Since the viral load in samples is determined as a result of the replication process in the human body after virus infection and within the patient's organs, this relationship is important. If you have data on the SARS-CoV-2 results from samples for each patient, please include them. If not, consider reviewing the relevant literature and discussing this in the manuscript :

We would like to express our appreciation for the insightful review.

In our study, the diagnosis of Covid-19 was established within our cohort through the utilization of a PCR test, which was not a quantitative test. Consequently, the quantitative analysis of the viral load within SARS-CoV-2 samples and its correlation with COVID-19 patient characteristics was regrettably not within the scope of our research.

While we acknowledge the significance of investigating viral load in relation to COVID-19 patients, our study was inherently limited by the tools and methodologies available for diagnosis. Nevertheless, this limitation emphasizes the need for further research that specifically incorporates quantitative viral load analysis to elucidate its potential implications in the context of COVID-19 patient management and outcomes. We appreciate the importance of this aspect and acknowledge its potential significance in the broader understanding of the disease.

Reviewer 2 Report

Comments and Suggestions for Authors

The present study investigates the prognostic significance of various markers of acute kidney injury (AKI) with regard to different outcomes (mortality, ICU admission, occurrence of acute kidney failure) in COVID-19 patients. Given the high rate of AKI in COVID-19 patients, the question has considerable clinical relevance. However, there are three points of improvement:

1. some formalities should be corrected in the manuscript:

- The term "urinary excretion rate" in the legends of figures 2A and 2B is incorrect, as an excretion rate describes the amount of a substance excreted per time (e.g. in mg/24h). For the markers LAP and CCP14, only their concentrations in the urine are given, from which no conclusions can be drawn about the actual amount of substance released. Rather, it should be mentioned as a limitation that the measured concentrations are significantly dependent on the degree of concentration of the urine (to be estimated e.g. from the ratio of creatinine in the urine to creatinine in the serum). If for example diuretics are administered, the urine is diluted and the concentration of the measured markers will drop, although the amount of released markers remains constant.

- In Figure 3, "U/g" is given as the unit for cystatin concentration in serum - a correct dimension of a concentration must be chosen here.

- In Table 2, starting with the row "Day 3", the contents of the columns are horizontally shifted against each other. This should be synchronized.

2. Primary endpoint is the "occurrence of AKI between Day 0 and Day 3 as defined according to AKIN criteria", but serum creatinine in the AKI group is significantly lower on day 3 (0.73 mg/dL, IQR 0.63-1.1) than on day 0 (0.93 mg/dL, IQR 0.72-1.1). Was the diagnosis of AKI based solely on the criterion of urinary output? There is a need for clarification here!

3. The T-test used also raises an important question: This test is explicitly to be used only for normally distributed variables (e.g. Kim TK, Korean Journal of Anesthesiology 2015;68(6):540-546), but all parameters are presented here as median and IQR. This is normally only the case for non-normally distributed parameters.

In the manuscript, no information is given on which test and which result was used to check the numerical parameters for normal distribution; this must be added.

Author Response

  1. some formalities should be corrected in the manuscript:

- The term "urinary excretion rate" in the legends of figures 2A and 2B is incorrect, as an excretion rate describes the amount of a substance excreted per time (e.g. in mg/24h). For the markers LAP and CCP14, only their concentrations in the urine are given, from which no conclusions can be drawn about the actual amount of substance released. Rather, it should be mentioned as a limitation that the measured concentrations are significantly dependent on the degree of concentration of the urine (to be estimated e.g. from the ratio of creatinine in the urine to creatinine in the serum). If for example diuretics are administered, the urine is diluted and the concentration of the measured markers will drop, although the amount of released markers remains constant.

We extend our sincere gratitude to the referee for his comprehensive review of our work. We acknowledge that there were issues with the terminology used in our manuscript, and we have taken his feedback into consideration. Specifically, we have made revisions to the figure legends, replacing "urinary excretion rate" by "urinary levels" for clarity purposes.

We concur with the referee's observation that the measured concentrations are reliant on creatinine levels. This interdependence is a crucial aspect to consider in our analysis.

The wide dispersion of urinary levels measured in our patient bgroups can be attributed to several factors. One prominent factor is the rapid onset of oligo-anuria in the AKI patient group. This condition, characterized by a significant reduction in urine output, can introduce variability in the measurements, as the kidneys may lose their capacity to effectively concentrate urine in such cases.

Furthermore, concerning urinary Leucine Aminopeptidase (LAP) results, the spectrofluorimetric assay measured the enzyme activity as compared to an internal standard requiring a calibration curve. Data were then expressed as a concentration and not an excretion rate. We acknowledge this represents a limitation to the interpretation of the results.

Regarding urinary CCL14, as per our methodology, we estimated its concentration based on the creatinine ratio. We made an error in the unit of measurement while constructing the figure, for which we offer our sincere apologies. Corrections have been made in the revised version of the manuscript.

- In Figure 3, "U/g" is given as the unit for cystatin concentration in serum - a correct dimension of a concentration must be chosen here.

We sincerely regret this oversight. The error in question was a result of an inadvertent oversight during the figure creation process. It is important to note that the unit of measurement has now been corrected to micrograms per milliliter (ug/ml). We appreciate your understanding and diligence in bringing this to our attention.

- In Table 2, starting with the row "Day 3", the contents of the columns are horizontally shifted against each other. This should be synchronized.

This has been arranged in proportion and synchronized in the revised version of the manuscript.

  1. Primary endpoint is the "occurrence of AKI between Day 0 and Day 3 as defined according to AKIN criteria", but serum creatinine in the AKI group is significantly lower on day 3 (0.73 mg/dL, IQR 0.63-1.1) than on day 0 (0.93 mg/dL, IQR 0.72-1.1). Was the diagnosis of AKI based solely on the criterion of urinary output? There is a need for clarification here!

We appreciate and acknowledge the insightful observation made by the reviewer, and we concur with the need for clarification on this matter. Indeed, the primary endpoint of our study was predicated on the "occurrence of AKI between Day 0 and Day 3" as defined in accordance with the Acute Kidney Injury Network (AKIN) criteria.

To provide further clarification, the diagnosis of AKI in our study was not based solely on the criterion of urinary output. Instead, it adhered to a multifactorial approach, encompassing assessments of both creatinine levels and urine output, in conjunction with the consideration of the necessity of renal replacement therapy.

We agree that the observed decrease in serum creatinine levels in the AKI group on Day 3 (0.73 mg/dL, IQR 0.63-1.1) as compared to Day 0 (0.93 mg/dL, IQR 0.72-1.1) may be attributed to the rapid development of oligo-anuria in AKI patients. Along these lines, the consecutive administration of substantial volumes of saline serum IV perfusion, as a part of clinical management, may have contributed to a degree of hemodilution.  Consequently, while creatinine levels did not exhibit a statistically significant change on Day 3, the primary basis for diagnosing AKI was related to the evident manifestation of oligo- and anuria in these patients, in accordance with AKIN criteria.

A paragraph discussing the matter has been added to the discussion.

  1. The T-test used also raises an important question: This test is explicitly to be used only for normally distributed variables (e.g. Kim TK, Korean Journal of Anesthesiology 2015;68(6):540-546), but all parameters are presented here as median and IQR. This is normally only the case for non-normally distributed parameters.

In the manuscript, no information is given on which test and which result was used to check the numerical parameters for normal distribution; this must be added.

We appreciate the reviewer's feedback and would like to address the concerns raised regarding the use of the t-test and the need to check for normal distribution in the manuscript.

In response to the reviewer's comment, we have revised our methodological approaches. We  used the Wilcoxon rank-sum test (Mann-Whitney U test) to compare the patient groups, without the need for normality checks. The Mann-Whitney U test is a non-parametric test that does not assume normality. These changes have been explained in the methodology section of the manuscript and the results were exactly the same .

We believe that this adjustment ensures a robust and appropriate statistical analysis for our data, and we thank the reviewer for their valuable input.

Reviewer 3 Report

Comments and Suggestions for Authors

I read with great interest the paper "Longitudinal Follow-Up of Dedicated Serum and Urine Biomarkers of COVID-19-Associated Acute Kidney Injury: Diagnostic and Prognostic Impacts"

Several biomarkers were evaluated such as NGAL, LAP, CCL14 and Cystatin C, offering insights into short-term morbidity and mortality risks in COVID-19 patients, regardless of AKI occurrence.

Overall, the manuscript is clear and well-written.

Did authors hypothesized to evaluate proteinuria as a predictor of AKI? Authors may include this in the limitations or in the discussion. please refer to  doi: 10.23736/S2724-6051.21.04308-1.

Author Response

We are grateful for the valuable feedback provided by the reviewer, and we are committed to addressing the concerns raised regarding the proteinuria measurements at the admission day. While we indeed collected the requested data, we inadvertently omitted to include them in the initial version of the manuscript. The rationale behind this omission is that, upon thorough analysis, no statistically significant difference in proteinuria levels was found between the two groups, namely AKI and non-AKI.

In response to the reviewer's suggestion, we have inserted the relevant data into Table 2 for comprehensive reference. Furthermore, we have added a dedicated paragraph to the discussion section of the revised manuscript to provide context and insight into the absence of a statistical distinction in proteinuria levels between the two groups (a recent review regrading this point has been added). We believe that this additional information will enrich the discussion.

Round 2

Reviewer 2 Report

Comments and Suggestions for Authors

All my points of criticism have been dealt with satisfactorily, in my view there are no further suggestions for improvement.